# FeNO 350 mL/s: Unlocking the Small Airways to Achieve Clinical Remission in Severe Asthma—A Pilot Study

**DOI:** 10.3390/arm93050037

**Published:** 2025-09-17

**Authors:** Vitaliano Nicola Quaranta, Andrea Portacci, Leonardo Maselli, Marta Tornesello, Maria Granito, Gennaro Rociola, Silvano Dragonieri, Giovanna Elisiana Carpagnano

**Affiliations:** Institute of Respiratory Disease, University of Bari, 70124 Bari, Italy; vitalianonicola.40@gmail.com (V.N.Q.); a.portacci01@gmail.com (A.P.); jaleomaselli@gmail.com (L.M.); martorne97@gmail.com (M.T.); mariagranito1998@gmail.com (M.G.); gen.rociola98@gmail.com (G.R.); elisiana.carpagnano@uniba.it (G.E.C.)

**Keywords:** FeNO 350 mL/s, small airways disease, severe asthma, clinical remission

## Abstract

**Highlights:**

**What are the main findings?**

**What is the implication of the main finding?**

**Abstract:**

**Background:** Several studies focused on the importance of managing small airways disease in the treatment of severe asthma, whose improvement can improve respiratory symptoms, lung function, and airways inflammation, potentially reaching the objective of clinical remission. **Methods:** Twenty-five patients with severe asthma and without bronchiectasis were enrolled. They were started on biological therapies with Omalizumab, Dupilumab, Benralizumab or Mepolizumab. Follow-up evaluations were conducted at baseline (T0) and after one year of biological therapy (T1). Assessments included clinical evaluations, spirometry, questionnaires, and inflammatory markers. **Results:** Predictive analysis identified baseline FeNO 350 mL/s levels as a significant predictor of clinical remission in both univariable and multivariable analysis. Higher FeNO 350 mL/s levels at T0 were associated with an increased likelihood of achieving remission (*p* = 0.012). The optimal cutoff value for FeNO 350 mL/s was determined to be 18 ppb, based on the Younden Index. **Conclusions:** Following patients with severe asthma on biological therapy for one year, FeNO 350 mL/s could be used as a predictive factor of clinical remission, highlighting its importance as inflammatory marker not only in small airways disease, but also in predicting clinical remission in severe asthmatic patients.

## 1. Introduction

Asthma is a highly prevalent chronic condition, affecting approximately 300 million individuals worldwide [1]. Although severe asthma accounts for less than 10% of the overall asthma population, it is responsible for a disproportionately high share of asthma-related healthcare costs. This condition imposes a significant economic burden on healthcare systems, with expenses rising notably in recent years. Therefore, the primary goals of asthma management are to achieve effective symptom control and minimize the risk of future exacerbations, ultimately preserving lung function [2].

In recent years, growing attention has been directed towards the role of small airways disease (SAD) and the concept of clinical remission. While small airways dysfunction is well recognized in asthma, its exact contribution to disease severity and control remains incompletely understood. The ATLANTIS study investigated the relevance and extent of small airways dysfunction by assessing combinations of biomarkers, physiological tests, and imaging markers to best measure SAD in asthma patients [3]. Interestingly, this study revealed that approximately 91% of individuals with asthma present with a phenotype characteristic of small airways disease [4], highlighting the need for targeted therapeutic approaches within the framework of precision medicine.

Effective asthma management thus increasingly focuses on controlling small airway involvement, particularly by identifying inflammatory markers associated with SAD, such as those assessed by impulse oscillometry (IOS) [5] and fractional exhaled nitric oxide (FeNO) measured at a flow rate of 350 mL/s.

Currently, the therapeutic goal in severe asthma is to achieve clinical remission. According to the Severe Asthma Network Italy (SANI) Delphi consensus, clinical remission in severe asthma is defined by the discontinuation of oral corticosteroids (OCS), absence of asthma-related symptoms, absence of exacerbations or acute attacks, and stable lung function. Stable lung function’ refers to the absence of a clinically relevant decline in FEV_1_ over a 12-month period, without significant exacerbations or the need for systemic corticosteroids. While the SANI Delphi consensus did not endorse specific thresholds [6], subsequent studies have operationalized remission using FEV_1_ ≥ 80% predicted or no more than a 100 mL decline from baseline/FEV_1_ above the lower limit of normal [7,8,9]. For pragmatic reasons, we hereafter apply a tolerance of ±10% of baseline FEV_1_, which encompasses smaller variations. These criteria provide a practical framework for assessing treatment efficacy and guiding therapeutic decisions in this patient population.

Although FeNO measured at 350 mL/s has emerged as a promising marker of distal airway inflammation, its potential role in predicting clinical remission remains underexplored. Therefore, this study aims to assess whether baseline FeNO 350 mL/s levels can serve as a reliable predictor of clinical remission in patients with severe asthma undergoing biological therapy, evaluated after one year of treatment. By examining this potential biomarker, we seek to elucidate its diagnostic accuracy and support a precision medicine approach for the optimal management of severe asthma.

## 2. Materials and Methods

### 2.1. Study Design

This retrospective observational study was conducted at the Severe Asthma Center of the Respiratory Diseases Unit, University Hospital Policlinico of Bari. Patient enrollment took place between 10 October 2022 and 25 March 2024, and included individuals diagnosed with severe asthma who were referred for biological therapy.

Each patient underwent two standardized evaluations during the study period. The first evaluation, referred to as T0, was conducted at baseline—immediately prior to the initiation of biological therapy. The second evaluation, labeled T1, was performed after 12 months of continuous treatment. At both time points, a comprehensive clinical, functional, and inflammatory assessment was conducted, allowing for comparison of pre- and post-treatment data.

Patients were eligible for inclusion if they were aged 18 years or older and had complete data available for both T0 and T1 visits. Importantly, only patients who underwent a full set of pulmonary function tests, including fractional exhaled nitric oxide at 350 mL/s (FeNO350), were included in the analysis. Patients with incomplete testing or missing FeNO350 were excluded. Consequently, no patients treated with Omalizumab or Tezepelumab fulfilled these criteria, and the final study population consisted exclusively of patients receiving Mepolizumab, Benralizumab, or Dupilumab

Patients were excluded if they had chronic pulmonary conditions that could confound the interpretation of small airways disease or FeNO measurements. Additional exclusion criteria included active smoking at the time of enrollment, incomplete data, or loss to follow-up before the 12-month evaluation (Figure 1).

In our center, patients starting biologic therapy undergo a standardized follow-up protocol. After the baseline visit (T0), a first clinical reassessment is performed after 3–4 months to evaluate early response and tolerability. All patients included in this study continued their prescribed biologic, as no non-responders were identified at this time. Patients are provided with direct contact (telephone and e-mail) to report any clinical issues at any time. If concerns emerge, an additional visit is arranged around 6–8 months; otherwise, patients proceed to the 12-month visit, where full functional, biomarker, and clinical reassessments are performed. Notably, at the intermediate outpatient visits no data were systematically recorded in the database and FeNO350 was not performed. For the purposes of this analysis, only standardized T0 and T1 evaluations were considered.

Although the study was retrospective, all patients were re-contacted by telephone and asked to provide written informed consent for the use of their clinical and functional data in this research; only those who returned signed consent forms were included in the analysis. The study was approved by the Ethics Committee of the A.O.U. Policlinico of Bari (protocol no. 6313, approved 4 March 2020) and conducted in accordance with the Declaration of Helsinki (Figure 1 illustrates the study enrollment and follow-up process).

### 2.2. Data Collection

During the 18-month enrollment period, a total of 56 patients initiated biologic therapy at our Severe Asthma Center. Twenty-five patients fulfilled all inclusion criteria (availability of paired baseline and 12-month follow-up data, including FeNO350 measurements) and were included in the final analysis. Thirty-one patients were excluded due to one or more of the following reasons: (i) absence of FeNO350 measurement at baseline and/or follow-up, either because of temporary malfunction of the system or inability to perform the maneuver with adequate coordination; (ii) missing 12-month follow-up. None of the excluded patients had FeNO350 available at both time points, which was an essential prerequisite for inclusion. A complete list of excluded patients, with biologic therapy, age, sex, and reason for exclusion, is provided in Appendix A.

For each patient, comprehensive data were collected at both time points, T0 and T1. These included demographic characteristics (age, sex, body mass index, smoking history), clinical history (asthma duration, age at diagnosis, presence of atopy), and associated comorbidities such as eosinophilic granulomatosis with polyangiitis (EGPA), eosinophilic pneumonia, hypereosinophilic syndrome, chronic rhinosinusitis, nasal polyps, urticaria, vocal cord dysfunction, gastroesophageal reflux disease (GERD), obstructive sleep apnea syndrome (OSAS), and anxiety–depressive disorders.

Clinical outcomes were assessed using validated tools including the Asthma Control Test (ACT) and the Asthma Control Questionnaire-6 (ACQ6). The number of exacerbations—both asthma-related and infectious—hospital admissions, unscheduled healthcare visits, and use of oral corticosteroids (including average annual dosage) were also recorded. Pulmonary function tests included spirometric measurements such forced expiratory volume in the first second (FEV1), forced vital capacity (FVC), the FEV1/FVC ratio (Tiffenau index), and maximal expiratory flow at 25% of FVC (MEF25). Inflammatory markers were evaluated through fractional exhaled nitric oxide (FeNO) measurements at 50 and 350 mL/s flow rates, alongside laboratory parameters including white blood cell counts, peripheral eosinophil counts, and total immunoglobulin E (IgE) levels.

The primary objective of the study was to evaluate whether baseline FeNO measured at 350 mL/s could serve as a predictive biomarker for achieving clinical remission, as defined by the Delphi consensus from the Severe Asthma Network Italy (SANI). According to this definition, clinical remission encompasses four key domains: (1) discontinuation of oral corticosteroids (OCS); (2) absence of asthma-related symptoms; (3) no exacerbations or acute attacks; and (4) stable lung function, defined as no clinically relevant decline in FEV_1_ over a 12-month period. In line with recent registry studies, we adopted a pragmatic definition of FEV_1_ ≥ 80% predicted or ≤100 mL decline from baseline, and for clinical consistency applied a tolerance of ±10% of baseline FEV_1_ [6,7,8,9]. This composite outcome was systematically assessed for each patient at the 12-month follow-up visit, based on both clinical evaluation and objective functional and inflammatory parameters.

### 2.3. Questionnaires

All patients in the study completed three questionnaires: the Asthma Control Test (ACT) [10], the Test of Adherence to Inhalers (TAI) [11], and the Asthma Control Questionnaire 6 (ACQ6) [12]. These tools were used to evaluate the level of asthma control and the extent of the adherence to the prescribed inhaler treatment regimen.

### 2.4. Lung Function

Respiratory function tests for the enrolled patients were performed using the Masterlab Jaeger spirometer, manufactured by CareFusion (Hoechberg, Germany), adhering to the European Respiratory Society/American Thoracic Society Standards [13].

### 2.5. Measurement of Exhaled Nitric Oxide (FeNO) 50 and 350 mL/s

FeNO levels were measured at two different flow rates, 50 mL/s and 350 mL/s, utilizing an eletrochemical-based analyzer, the FeNO+ by Medisoft-MGCD, Saint Paul, MN, USA. The measurement process was conducted in accordance with the manufacturer’s instructions and followed the guidelines recommended by the European Respiratory Society [14].

To accurately assess FeNO levels while eliminating the influence of nasal NO, a specialized breathing method was used. This method involved expiratory resistance and positive pressure within the month to maintain a steady expiratory flow at both 50 and 350 mL/s [15].

### 2.6. Impulse Oscillometry (IOS)

Impulse oscillometry was performed using the MasterScreen™ IOS system (Jaeger, Hoechberg, Germany), in accordance with the European Respiratory Society (ERS) recommendations for the forced oscillation technique. The procedure consisted of quiet tidal breathing for approximately 30 s, during which the operator stabilized the patient’s cheeks to minimize upper airway interference.

The following parameters were recorded:R5–R20 (kPa·s·L^−1^): This represents the difference in resistance between 5 Hz and 20 Hz, serving as a surrogate marker for peripheral (small) airway resistance. Values exceeding 0.07 kPa·s·L^−1^ are indicative of small airways dysfunction (SAD) [16].X5 (kPa·s·L^−1^): Reactance at 5 Hz, reflecting the elastic properties of the distal lung; more negative values suggest reduced compliance and increased stiffness of the peripheral airways.Fres (Hz): Resonant frequency at which the reactance equals zero, representing the balance point between capacitive and inertial properties of the respiratory system.AX (kPa/L): The area under the reactance curve from 5 Hz to Fres, used as an index of the overall load on the peripheral airways and lung compliance [17].

### 2.7. Statistical Analysis

All statistical analyses were conducted using SPSS v.23 (IBM Corp., Armonk, NY, USA). The Kolmogorov–Smirnov test was performed to assess the normality of continuous variables. Given that the data did not follow a normal distribution, non-parametric tests were employed for further analyses. Continuous variables were expressed as medians with interquartile ranges (IQR), while categorical variables were presented as frequencies and percentages. For inter-group comparisons, the Mann–Whitney U test was used for continuous variables, while categorical variables were compared using the chi-square test or Fisher’s exact test.

A formal sample size calculation was not performed because FeNO measured at 350 mL/s represents a novel predictive parameter with no prior studies available to provide reference values for estimation. Moreover, the retrospective design of the present study makes prospective sample size calculation not applicable.

To evaluate changes over time within each group, the Wilcoxon signed-rank test was used to compare paired continuous variables at baseline (T0) and after one year of follow-up (T1).

Logistic regression analysis was conducted to identify predictors of clinical remission. Odds ratios (ORs) and their 95% confidence intervals (CIs) were reported for each variable. Variables that emerged as significant predictors in the univariable analysis were entered into the multivariable logistic regression model, together with the main potential confounding factors such as sex and age.

The predictive accuracy of FeNO350 for clinical remission was assessed using a receiver operating characteristic (ROC) curve analysis. The area under the curve (AUC) was calculated, and the optimal cutoff point was determined using the Youden Index to maximize sensitivity and specificity.

Significance levels were set at α < 0.050.

## 3. Results

### 3.1. Baseline Characteristics of the Study Population

The study included 25 patients with severe asthma who had no bronchiectasis. The median age of the cohort was 51 years (IQR: 41–59), and 44% were women. The median BMI was 26 kg/m^2^ (IQR: 24–30), with the majority of patients being non-smokers (64%) and a smaller proportion being past smokers (36%). The median duration of asthma was 20 years (IQR: 3–38), with a median age of diagnosis at 23 years (IQR: 15–44). Common comorbidities included chronic rhinosinusitis and nasal polyposis, each present in 56% of the population, as well as gastroesophageal reflux disease (GERD), observed in 44% of cases (Table 1).

At baseline, subgroup analysis by biologic therapy revealed differences in inflammatory markers and corticosteroid exposure. The prescription of each biologic was performed according to the official criteria of the Italian Medicines Agency (AIFA). Patients treated with mepolizumab had a median eosinophil count of 490 (383–575) cells/µL, FeNO50 of 20 (13–67) ppb, and 1.5 (0.25–4.25) OCS cycles. In the benralizumab group, median eosinophils were 600 (438–803) cells/µL, FeNO50 12 (9–20) ppb, and OCS cycles 1.0 (1–3.25). Patients receiving dupilumab showed lower blood eosinophils [260 (232–284) cells/µL] but higher FeNO50 [26 (24–34) ppb], with OCS cycles 1.0 (1–4).

The prescription of each biologic was performed according to the official criteria of AIFA [18]

### 3.2. Inter-Group Comparisons at Baseline (T0)

At baseline (T0), the population was divided into two groups: patients with uncontrolled asthma (n = 14) and those who achieved clinical remission (n = 11) after one year of follow-up. At T0, there were no statistically significant differences in age, sex, or BMI between the two groups, although trends were observed. For instance, the remission group tended to have a lower BMI (23.5 vs. 27.5 kg/m^2^) and a lower percentage of female participants (27.3% vs. 57.1%). Comorbidities such as chronic rhinosinusitis and nasal polyposis were more prevalent in the remission group (72.7% vs. 42.9%), though these differences were not statistically significant. Interestingly, baseline fractional exhaled nitric oxide at 350 mL/second (FeNO350) levels were significantly higher in the remission group compared to the uncontrolled asthma group (13.5 vs. 10 ppb, *p* = 0.012), suggesting a potential link to treatment response. Additionally, a non-significant trend toward lower resonance frequency (Fres) was observed in the remission group (16 Hz vs. 19 Hz, *p* = 0.061) (Table 2).

### 3.3. Inter-Group Comparisons After One Year (T1)

After one year of treatment (T1), clinical outcomes between the two groups diverged. Patients in the clinical remission group demonstrated significant improvements in asthma control, as reflected by a reduction in the Asthma Control Questionnaire (ACQ6) score from a baseline of 2.57 to 0.81 (*p* = 0.005). This contrasted with the uncontrolled asthma group, which showed no significant change in ACQ6 scores over time. Lung function, as measured by FEV1%, improved in both groups, with the remission group achieving a higher median value (91% vs. 71.5%), although this difference did not reach statistical significance. Importantly, FeNO350 levels decreased in both groups at T1, with the remission group maintaining slightly lower levels, but these changes were not statistically significant at follow-up.

### 3.4. Intra-Group Comparisons over Time

When comparing changes within each group over time, distinct patterns emerged. In the clinical remission group, significant improvements were observed in asthma control metrics, such as ACQ6 and ACT scores, along with a reduction in FeNO350 levels. In contrast, the uncontrolled asthma group showed minimal improvement in clinical outcomes, with only slight, non-significant reductions in exacerbations and corticosteroid use.

#### Subgroup Analysis: Benralizumab-Treated Patients

Among patients with severe asthma treated with biologics, baseline FeNO measured at 350 mL/s (FeNO350) was significantly higher in those who achieved clinical remission compared with uncontrolled patients. After one year, remission patients showed marked improvements in asthma control (lower ACQ6, higher ACT), absence of exacerbations, and reduced type-2 inflammatory markers, while spirometric indices improved without reaching statistical significance. Subgroup analysis restricted to benralizumab-treated patients confirmed these trends, highlighting that higher baseline FeNO350 identified patients more likely to achieve remission. Overall, FeNO350 emerged as a robust predictor of clinical remission, with an optimal cutoff of 18 ppb showing high specificity and good diagnostic accuracy (Appendix A).

### 3.5. Predictive Value of FeNO350

Predictive analysis identified baseline FeNO350 levels as a significant predictor of clinical remission in both univariate and multivariate analyses. In the univariate analysis, higher FeNO350 levels at T0 were associated with an increased likelihood of achieving remission, with an odds ratio of 1.115 (95% CI: 1.008–1.234, *p* = 0.035) (Figure 2).

In the multivariate analysis, after adjusting for principal confounding factors such as age, sex, and BMI, FeNO350 levels remained a significant predictor, with an adjusted odds ratio of 1.143 (95% CI: 1.007–1.298, *p* = 0.039). This suggests that higher baseline FeNO350 levels independently predict clinical remission. Other variables, including age, BMI, and comorbidities such as GERD, nasal polyposis, and chronic rhinosinusitis, did not show significant predictive value in the adjusted model (Table 3).

### 3.6. Diagnostic Accuracy of FeNO350

FeNO350 levels were analyzed using a newly identified cutoff value of 18 ppb to differentiate patients achieving complete remission from those not in remission. The sensitivity of FeNO350 at this cutoff was 63.6%, indicating its ability to correctly identify patients in remission. The specificity was 92.8%, reflecting its accuracy in correctly classifying patients not in remission. The positive predictive value (PPV) was 87.5%, representing the likelihood that patients predicted to be in remission were correctly identified, while the negative predictive value (NPV) was 76.4%, indicating the probability that patients not predicted to be in remission were accurately classified.

The diagnostic accuracy of FeNO350 for predicting remission was evaluated using a ROC curve. The analysis revealed an area under the curve (AUC) of 0.799 (*p* = 0.012), indicating good predictive performance. The optimal cutoff value for FeNO350 was determined to be 18 ppb, based on the Youden Index. In addition, we also evaluated other cutoff values (10, 14, 20, and 24 ppb) to explore their diagnostic accuracy, but 18 ppb provided the best balance between sensitivity and specificity (Figure 3, Table 4).

## 4. Discussion

Our study provides novel and clinically impactful evidence demonstrating that FeNO measured at 350 mL/s is a strong predictor of clinical remission in severe T2-high asthma patients treated with biologics. Specifically, we identified a cutoff value of 18 ppb, above which patients were significantly more likely to achieve remission, offering a practical and easily applicable biomarker to guide treatment decisions and personalize therapy.

FeNO at 350 mL/s specifically reflects nitric oxide production from the small peripheral airways, unlike the standard 50 mL/s measurement that primarily captures central airway inflammation. The physiological basis lies in the expiratory flow dependence of FeNO: as flow increases, sampling shifts distally, capturing NO generated in smaller bronchioles [19]. This finding is particularly relevant given that small airways constitute over 98% of the lung surface [20], and the so-called “silent zone” (<2 mm in diameter) significantly contributes to airway resistance. Their reaction to irritants leads to damage and structural remodeling, altering elasticity and impacting airflow dynamics [21]. SAD affects 50–60% of individuals with asthma [22] and remains underrecognized by standard spirometry, which poorly detects distal airflow limitation.

The ATLANTIS study involving 773 patients reported a SAD prevalence of 91%, emphasizing the importance of assessing small airway involvement in asthma severity and control [3,4]. Our study adds to this by proposing FeNO 350 mL/s as a feasible biomarker reflecting distal airway inflammation. Elevated FeNO is known to correlate with type 2 eosinophilic inflammation [23,24], and previous studies have shown that higher FeNO 50 mL/s predicts treatment response and improved outcomes in severe asthma [15,25,26]. Pianigiani et al. [27] confirmed that FeNO 50 mL/s decreases significantly under biologic therapy, particularly with benralizumab and dupilumab, suggesting FeNO’s utility in guiding biologic therapy.

However, FeNO 350 mL/s offers additional value by specifically assessing small airway inflammation. Chan and Lipworth [19] reviewed the impact of biologics on SAD, showing improvements in small airway indices with systemic biologics such as anti-IL5, anti-IL4Rα, anti-TSLP, and anti-IgE therapies, which reach distal airways unlike inhaled therapies limited by deposition characteristics. Abdo et al. demonstrated that baseline SAD measures, particularly R5–R20, predicted clinical response to biologics better than blood eosinophils or FeNO 50 mL/s, highlighting SAD as a distinct treatable trait [28]. Conversely, Chan and Lipworth [29] showed that while benralizumab improved FEV1, it did not significantly change oscillometry-measured SAD parameters, suggesting that structural changes may limit reversibility despite reduced inflammation. Our finding that FeNO350 predicts remission emphasizes the importance of targeting active distal eosinophilic inflammation, which may remain modifiable even in the presence of airway remodeling.

In our study, impulse oscillometry confirmed that most patients at baseline (T0) exhibited small airways dysfunction, with elevated R5–R20 values above 0.07 kPa·s·L^−1^. This is consistent with the previous literature, including the ATLANTIS study which reported a prevalence of SAD in over 90% of asthma patients. However, in our cohort, SAD alone did not discriminate between patients who achieved clinical remission and those who did not. Notably, only elevated FeNO at 350 mL/s—a marker of distal airway inflammation—was significantly associated with remission. This finding supports the hypothesis raised by Abdo et al., who suggested that active eosinophilic inflammation in the small airways, rather than structural alterations per se, is the key driver of treatment response. In addition, although non-responders tended to present with later asthma onset, longer disease duration, higher BMI, and a female predominance, these features did not reach statistical significance nor independently predict remission in multivariate analysis. Taken together, these results emphasize that demographic and anthropometric characteristics may shape the severe asthma phenotype, but small airway inflammation—as reflected by FeNO350—remains the most relevant determinant of clinical remission. Thus, while SAD appears to be a common trait in severe asthma, it is the presence of active, reversible inflammation—as captured by FeNO350—that may best predict the potential for clinical remission.

Moreover, the significant reduction in FeNO350 levels from T0 to T1 observed in the clinical remission subgroup raises the intriguing possibility that this parameter may not only serve as a predictor, but also as a potential marker of “inflammatory remission” in the small airways. Although this remains speculative, it opens a new line of investigation: can FeNO350 be used to monitor the resolution of distal airway inflammation over time? If so, FeNO350 could evolve into a dynamic biomarker capable of identifying patients who achieve not only clinical but also inflammatory remission under biologic therapy.

Overall, these studies, combined with our results, indicate that FeNO 350 mL/s ≥ 18 ppb is a practical, non-invasive, and effort-independent biomarker to identify patients with active distal airway inflammation who are more likely to achieve remission with biologic therapy. This aligns with precision medicine goals to identify treatable traits and tailor therapies to individual patient profiles for optimal outcomes.

This study has several limitations. The sample size was relatively small, limiting generalizability, and its retrospective design may introduce selection bias and unmeasured confounding. The follow-up period was limited to one year, and the longer-term sustainability of remission remains unknown. For these reasons, we are extending the follow-up period and increasing sample size, including patients initiating tezepelumab therapy, to further validate these findings. Given that four different biologics were employed (omalizumab, mepolizumab, benralizumab, and dupilumab), variability in their mechanisms of action should be acknowledged as a limitation, as this may have affected certain results. Another limitation relates to the structure of the follow-up and data collection. Although patients were clinically reassessed at 3–4 months after initiating biologic therapy and none were classified as non-responders, information from these visits was not systematically entered into the study database, and FeNO350 was not performed at that time. Similarly, an additional visit could be offered at 6–8 months in case of clinical concerns, but only baseline and 12-month evaluations were standardized and included complete functional and biomarker data. While this reflects real-life practice in our center, a more granular follow-up with systematic collection of intermediate data might have provided further insights into the dynamics of treatment response.

Future research should include larger, prospective multicenter studies to validate FeNO350 ≥ 18 ppb as a biomarker of clinical remission. Mechanistic studies are needed to elucidate how FeNO350 reduction correlates with distal airway inflammation resolution and structural remodeling. Evaluating FeNO350 in relation to newer biologics targeting upstream alarmins (e.g., tezepelumab) could further expand its clinical utility.

## 5. Conclusions

In conclusion, FeNO 350 mL/s ≥ 18 ppb emerges as a promising, clinically relevant biomarker reflecting small airway inflammation and predicting clinical remission in severe T2-high asthma patients treated with biologics. Its integration into routine assessment could support precision medicine approaches, optimizing biologic therapy selection and improving outcomes by targeting distal airway inflammation, a key driver of disease burden in severe asthma. Further large-scale studies are warranted to confirm these findings and establish FeNO350 as a standard tool in the management of severe asthma.

## Figures and Tables

**Figure 1 arm-93-00037-f001:**
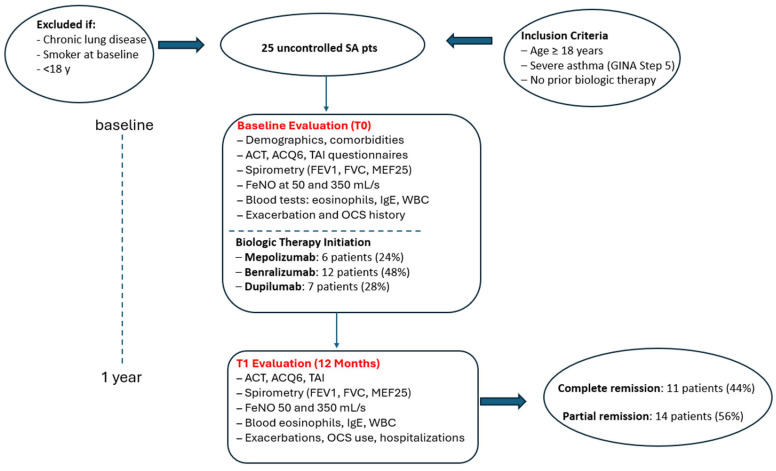
Study design and outcomes flowchart. Abbreviation list: SA: Severe Asthma; GINA: Global Initiative for Asthma; OCS: Oral Corticosteroids; ACT: Asthma Control Test; ACQ6: Asthma Control Questionnaire—six items; TAI: Test of Adherence to Inhalers; FeNO: Fractional exhaled Nitric Oxide; FEV1: Forced Expiratory Volume in 1 s; FVC: Forced Vital Capacity; MEF25: Maximal Expiratory Flow at 25% of FVC; IgE: Immunoglobulin E; WBC: White Blood Cell count; T0: baseline assessment; T1: follow-up assessment at 12 months.

**Figure 2 arm-93-00037-f002:**
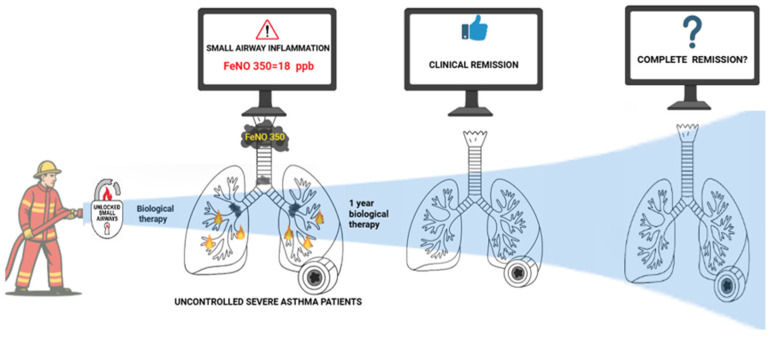
The firefighter symbolizes the initiation of biological therapy directed at uncontrolled severe asthma patients with small airway inflammation, indicated by elevated FeNO350 ≥ 18 ppb. The broken lock labeled “Unlocked Small Airways” represents the disruption of inflammation in the distal airways. Following one year of treatment, a subset of patients achieves clinical remission (middle panel). The final panel introduces the open question of whether this clinical improvement also reflects an “inflammatory remission” at the level of the small airways, as suggested by the decline in FeNO350. Abbreviations: FeNO, fractional exhaled nitric oxide; ppb, parts per billion; SAD, small airways disease.

**Figure 3 arm-93-00037-f003:**
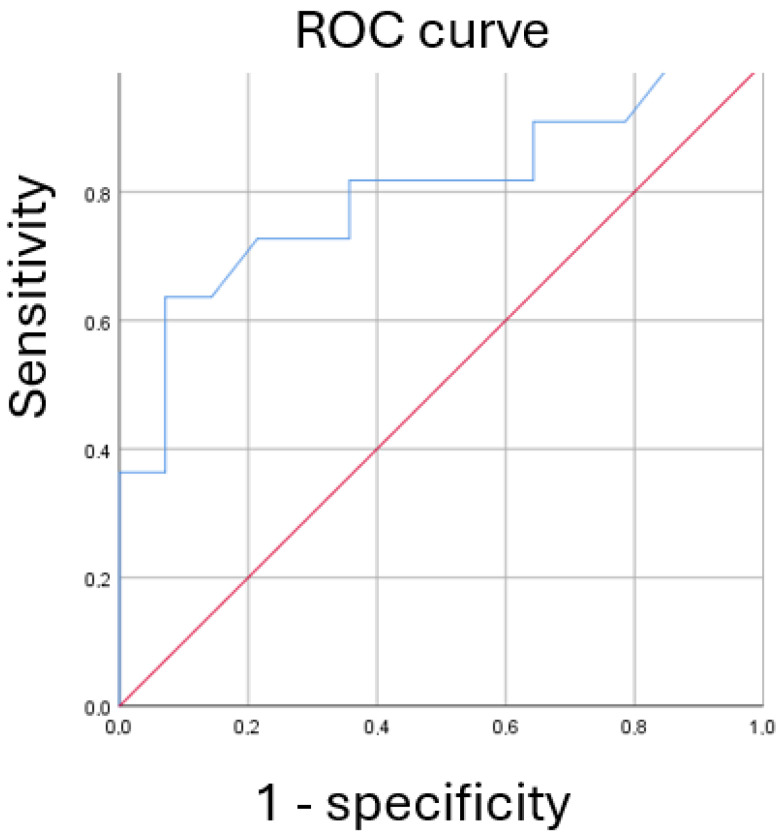
ROC curve for FeNO350 predicting complete remission in SA patients. Abbreviation List: ROC: Receiver Operating Characteristic; FeNO350: Fractional Exhaled Nitric Oxide at 350 mL/s; Sensitivity: true positive rate; 1−Specificity: false positive rate; Pts: Patients; SA: Severe Asthma. AUC = 0.799 (0.612; 0.985); *p* = 0.012; Youden Index: FeNO350 18 ppb.

**Table 1 arm-93-00037-t001:** Baseline characteristics and clinical features in uncontrolled asthma vs. clinical remission groups.

Parameters	Population N = 25	Uncontrolled Asthma N = 14	Clinical Remission N = 11	*p*-Value
Age y	51 (41; 59)	51.5 (43.5; 59)	38.5 (21; 56)	0.848
Sex f	11 (44)	8 (57.1)	3 (27.3)	0.138
BMI kg/m^2^	26 (24; 30)	27.5 (24; 30)	23.5 (21; 26)	0.144
History of smoke n (%)			0.325	
Current	0 (0)	0 (0)		
Past	9 (36)	5 (45.5)		
No	16 (64)	6 (54.5)		
Duration of asthma y median (IQ 25; 75)	20 (3; 38)	20(8.5; 39)	11(1; 21)	0.297
Age at asthma diagnosis y median (IQ 25; 75)	23 (15; 44)	23 (12.5; 45)	2.5 (1; 4)	0.247
EGPA n (%)	3 (12)	1 (7.1)	2 (18.2)	0.407
Eosinophilic Pneumonia n (%)	0 (0)			
Bronchiectasis n (%)	0 (0)			
Hypereosinophilic Syndrome n (%)	1 (4)	0 (0)	1 (9.1)	0.440
Chronic_Rhinosinusitis n (%)	14 (56)	6 (42.9)	8 (72.7)	0.138
Nasal Poliposis n (%)	14 (56)	6 (42.9)	8 (72.7)	0.138
Urticaria n (%)	1 (4.0)	1(7.1)	0 (0)	0.560
Vocal Cord Dysfunction n (%)	0 (0; 0)			
GERD n (%)	11 (44)	7 (50)	4(36.4)	0.393
OSAS n (%)	3 (12)	2 (14.3)	1 (9.1)	0.593
Anxiety n (%)	5 (20)	4 (28.6)	1 (9.1)	0.245

Abbreviations: N: Number of subjects; y: Years; Sex f: Female sex; BMI: Body Mass Index (kg/m^2^); EGPA: Eosinophilic Granulomatosis with Polyangiitis; GERD: Gastroesophageal Reflux Disease; OSAS: Obstructive Sleep Apnea Syndrome.

**Table 2 arm-93-00037-t002:** Comparison of Clinical and Functional Parameters in Uncontrolled Asthma vs. Clinical Remission at Baseline and After One Year.

	Population N = 25	T0 Baseline			T1 After 1 Year N = 25		
		Uncontrolled Asthma Pt N = 14	Clinical Remission Pt N = 11	*p*-Value	Uncontrolled Asthma Pt N = 14	Clinical Remission Pt N = 11	*p*-Value
Biological Therapy n (%)							0.140
Mepolizumab	6 (24)				2 (14.3)	4 (36.4)	
Bernalizumab	12 (48)				6 (42.9)	6 (54.5)	
Dupilumab	7 (28)				6 (42.9)	1.(9.1)	
Exacerbations (y) median (IQ 25; 75)	2.5 (1; 5)	* 2.5 (1; 5.5)	° 2.5 (1; 4)	0.283	0 (0; 1)	0 (0; 0)	0.058
Hospitalizations (y) median (IQ 25; 75)	0 (0; 1)	0 (0; 0.5)	0.5 (0; 1)	0.783	0 (0; 0)	0 (0; 0)	1
Infectious exacerbations (y) median (IQ 25; 75)	0 (0; 1)	0 (0; 0)	1 (1; 1)	0.430	0 (0; 0)	0 (0; 0)	0.200
Unscheduled visits (y) median (IQ 25; 75)	1 (0; 2)	* 1 (0; 2)	0.5 (0; 1)	0.117	0 (0; 0)	0 (0; 0)	1
OCS (y) median (IQ 25; 75)	2.5 (1; 5)	* 2.5 (1; 5.5)	° 2.5 (1; 4)	0.236	0 (0; 0)	0 (0; 0)	0.109
Average annual corticosteroid dosage (mg) median (IQ 25; 75)	17.5 (0.07; 0.34)	0.175 (0.07; 0.37)	0.17 (0.07; 0.27)				
ACT median (IQ 25; 75)	14 (10; 17)	* 12 (9.50; 16.50)	° 19.50 (16; 23)	0.100	23.50 (22; 25)	25 (23; 25)	0.295
ACQ6 median (IQ 25; 75)	2.20 (1.45; 3)	2.21 (1.43)	° 2.57	0.681	2.15 (1.12; 3.21)	0.81 (0.73; 0.88)	0.005
TAI median (IQ 25; 75)	54 (54; 54)	54 (54; 54)	54 (54; 54)	1	54 (54; 54)	54 (54; 54)	1
% FEV1 median (IQ 25; 75)	65 (55; 86)	65 (56; 82)	° 70.5 (55; 86)	0.742	71.50 (58; 97)	91 (64.5; 98)	0.642
FEV1 (L) median (IQ 25; 75)	1.98 (1.55; 2.41)	1.88 (1.51; 2.23)	° 2.41 (1.99; 2.84)	0.396	2.06 (1.85; 2.58)	3.10 (1.94; 3.57)	0.112
% FVC median (IQ 25; 75)	84.50 (75; 97)	84.50 (74.5; 97)	° 86 (75; 97)	0.742	92.50 (80; 104)	101 (82; 103)	0.681
FVC (L) median (IQ 25; 75)	2.93 (2.63; 4.30)	2.81 (2.53; 3.20)	° 3.64 (2.38; 4.58)	0.155	2.99 (2.51; 3.76)	3.88 (3.45; 4.73)	0.080
% FEV1/FVC median (IQ 25; 75)	81.50 (71; 85)	81.50 (71; 85)	78 (71; 85)	0.565	85 (70; 94)	90 (68; 97)	0.848
% REV median (IQ 25; 75)	1 (0; 3)	1 (0; 3.5)	1 (0; 2)	0.595	2 (2; 3)	3 (2; 3)	0.860
R5-20 kPa·L^−1^·s^−1^ IQ (25; 75)	0.10 (0.09; 0.13)	0.09 (0.07; 0.11)	0.11 (0.10; 0.15)	0.123	0.1 (0.07; 0.11)	0.12 (0.08; 0.16)	0.322
Fres Hz IQ (25; 75)	17 (15; 19)	16 (14; 18)	19 (16; 19)	0.061	18 (15; 19)	17 (15; 20)	0.956
AX kPa/L IQ (25; 75)	1.10 (0.5; 1.3)	1.30 (1.0; 1.4)	0.60 (0.5; 1.1)	0.297	1.20 (0.9; 1.4)	0.90 (0.6; 1.1)	0.334
X5 kPa·L^−1^·s IQ (25; 75)	−0.12 (−0.20; −0.11)	−0.12 (−0.20; −0.11)	−0.12 (−0.16; −0.12)	0.657	−0.14 (−0.18; −0.11)	−0.21 (−0.85; −0.11)	0.087
% MEF25 median (IQ 25; 75)	25 (15; 34)	25 (15; 33.5)	26 (16; 36)	0.443	34.5 (18; 50)	27 (16.5; 65.5)	0.742
FeNO50 (ppb) median (IQ 25; 75)	20 (8; 27)	23.50 (6.50; 30.50)	26 (16; 36)	0.476	17 (7; 39)	23 (17.5; 26.5)	0.956
FeNO350 (ppb) median (IQ 25; 75)	10 (5; 17)	10 (4.5; 16.5)	° 13.5 (8; 19)	0.012	10 (4; 18)	9 (6.5; 9.5)	0.459
WBC (cells/μL) median (IQ 25; 75)	8170 (7250; 9300)	8185 (716; 9510)	7710 (7250; 8170)	0.547	7105 (5800; 10,510)	7690 (6095; 8125)	0.956
EOS (cells/μL) median (IQ 25; 75)	389.3 (232; 700)	284 (231; 638)	° 600 (500; 700)	0.198	185 (0; 470)	0 (0; 40)	0.084
% EOS median (IQ 25; 75)	4.36 (3.2; 7.9)	4.10 (2.9; 7.33)	° 7.73 (6.90; 8.75)	0.080	2.61 (0; 6.61)	0 (0; 0.6)	0.170
IgE (cells/μL) median (IQ 25; 75)	248.50 (48; 485)	230 (42.4; 476.5)	600 (221; 980)	0.273	#		

* = comparison of uncontrolled asthma subgroup between T0 and T1 time: *p* value < 0.050; ° = comparison of complete remission subgroup between T0 and T1 time: *p* value < 0.050; # = missing value. Abbreviation List: N: Number of participants, T0: baseline, T1: after one year, Pt: Patients, y: Years, OCS: Oral Corticosteroids, ACT: Asthma Control Test, ACQ6: Asthma Control Questionnaire 6-item, TAI: Test of Adherence to Inhalers, % FEV1: Percentage of Forced Expiratory Volume in 1 s, FEV1 (L): Forced Expiratory Volume in liters, % FVC: Percentage of Forced Vital Capacity, FVC (L): Forced Vital Capacity in liters, % FEV1/FVC: Percentage of FEV1 to FVC ratio, % REV: Percentage of Reversibility, % MEF25: Maximum Expiratory Flow at 25% of FVC, R5–R20, resistance difference between 5 and 20 Hz; Fres, resonance frequency; AX, reactance area; X5, reactance at 5 Hz; FeNO50: Fractional Exhaled Nitric Oxide at 50 mL/s, FeNO350: Fractional Exhaled Nitric Oxide at 350 mL/s, WBC: White Blood Cell count, EOS: Eosinophils count, % EOS: Percentage of Eosinophils, IgE: Immunoglobulin E.

**Table 3 arm-93-00037-t003:** Logistic regression analysis of factors associated with clinical remission in asthma.

	Univariable Analysis			Multivariable Analysis		
Parameters	OR	%CI 95	*p*-value	OR	%CI	*p*-value
Age y	0.993	0.929–1.062	0.841	0.932	0.843–1.031	0.932
Sex f	0.281	0.052–1.536	0.222	0.322	0.036–2.907	0.312
BMI kg/m^2^	0.833	0.651–1.066	0.147			
History of smoking Current Former No	0.480	0.091–2.523	0.386			
Years of asthma y	0.96	0.916–1.018	0.204			
Age at asthma onset y	1.035	0.979–1.094	0.220			
Chronic Rhinosinusitis	3.282	0.574–18751	0.181			
Nasal Poliposis	3.556	0.651–19412	0.143			
GERD	0.571	0.114–2.872	0.497			
OSAS	0.600	0.047–7.630	0.694			
Anxiety	0.250	0.024–2.648	0.250			
Biological Therapy Omalizumab Mepolizumab Bernalizumab Dupilumab	0.301	0.082–1.101	0.070			
Exacerbations (y)	0.787	0.517–1.184	0.245			
Hospitalizations (y)	0.600	0.128–3.392	0.619			
Infectious exacerbations (y)	2.250	0.304–16.632	0.427			
Unscheduled visits (y)	0.531	0.218–1.297	0.531			
OCS (y)	0.749	0.483–1.160	0.195			
Average annual corticosteroid dosage (mg)	0.014	0.000–8.634	0.192			
ACT	1.144	0.970–1.347	0.110			
ACQ6	1.163	0.603–2.243	0.653			
% FEV1	0.995	0.962–1.030	0.784			
FEV1 (L)	1.770	0.631–4.965	0.278			
% FVC	0.988	0.941–1.037	0.619			
FVC (L)	2.005	0.753–5.341	0.164			
% FEV1/FVC	0.987	0.935–1.041	0.210			
% REV	1.110	0.981–1.382	0.351			
% MEF25	1.004	0.97–1.033	0.759			
FeNO50 (ppb)	1.018	0.986–1.051	0.264			
FeNO350 (ppb)	1.115	1.008–1.234	0.035 *	1.143	1.007–1.298	0.039 *
WBC (cells/μL)	1.000	0.999–1.000	0.657			
EOS (cells/μL)	1.001	0.999–1.004	0.228			
% EOS	1.203	0.954–1516	0.119			
IgE (cells/μL)	1.003	0.998–1.008	0.215			

Abbreviation List: OR: Odds Ratio; % CI 95: 95% Confidence Interval; *p* value: statistical significance level; AGE y: age in years; SEX f: female sex; BMI Kg/m^2^: Body Mass Index; GERD: Gastroesophageal Reflux Disease; OSAS: Obstructive Sleep Apnea Syndrome; OCS: Oral Corticosteroids; ACT: Asthma Control Test; ACQ6: Asthma Control Questionnaire (6-item); % FEV1: Percentage of Forced Expiratory Volume in 1 s; FEV1 (L): Forced Expiratory Volume in liters; % FVC: Percentage of Forced Vital Capacity; FVC (L): Forced Vital Capacity in liters; % FEV1/FVC: Percentage of FEV1 to FVC ratio; % REV: Percentage of Reversibility; % MEF25: Maximum Expiratory Flow at 25% of FVC; FeNO50 (ppb): Fractional Exhaled Nitric Oxide at 50 mL/s; FeNO350 (ppb): Fractional Exhaled Nitric Oxide at 350 mL/s; WBC (cells/μL): White Blood Cell count; EOS (cells/μL): Eosinophil count; % EOS: Percentage of Eosinophils; IgE (cells/μL): Immunoglobulin E; *: *p* < 0.050.

**Table 4 arm-93-00037-t004:** Diagnostic accuracy of FeNO350 in detecting complete remission.

Cutoff	Sensitivity	Specificity	PPV	NPV
18	63.6 (35.4; 84.8)	92.8 (68.5; 98.7)	87.5 (52.9; 97.7)	76.4 (90.4; 52.7)
10	81.8 (48.2–97.7)	64.3 (35.1–87.2)	64.3 (31.6–86.1)	81.8 (48.2–97.7)
14	72.7 (39.0–94.0)	64.3 (35.1–87.2)	61.5 (31.6–86.1)	75.0 (42.8–94.5)
20	54.5 (23.4–83.3)	92.9 (66.1–99.8)	85.7 (42.1–99.6)	72.2 (46.5–90.3)
24	36.4 (10.9–69.2)	92.9 (66.1–99.8)	80.0 (28.3–99.5)	65.0 (40.8–84.6)

Abbreviation List: Sensitivity: true positive rate; Specificity: true negative rate; PPV: Positive Predictive Value; NPV: Negative Predictive Value; FeNO350: Fractional Exhaled Nitric Oxide at 350 mL/s.

## Data Availability

Dataset is available upon reasonable request.

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
