# Peer review of "FeNO 350 mL/s: Unlocking the Small Airways to Achieve Clinical Remission in Severe Asthma—A Pilot Study"

_arm, 2025, doi:10.3390/arm93050037_

Round 1

Reviewer 1 Report

Comments and Suggestions for Authors

Thank you for providing me the opportunity to review this manuscript. In general, this is an interesting article. The authors aimed to assess whether baseline FeNO 350 mL/s levels can serve as a reliable predictor of clinical remission in patients with severe asthma undergoing biological therapy, evaluated after one year of treatment. They found that FeNO 350 ml/s could be used as a predictive factor of clinical remission, highlighting its  importance as inflammatory marker not only in small airways disease, but also in predicting clinical remission in severe asthmatic patients. However, there are some points should be improved.

Comments  

Title

            Due to small sample size, thus the “pilot study” should be added to the title.

Abstract

  1. Line 26, “wew” must be change to “was”.

 Introduction

  1. Line 64, “forced expiratory volume in one second (FEV1)” should be changed to “forced expiratory volume in the first second (FEV1)”.

Methods

  1. Line 93, “All patients signed written informed consent.” Due to the retrospective observational study. Please clarify the method that you use for signing inform consent.
  2. Line 110, “forced expiratory volume in one second (FEV1)” should be changed to “forced expiratory volume in the first second (FEV1)”.
  3. Line 152-153, Values exceeding 0.07 kPa·s·L⁻¹ are indicative of small airways dysfunction (SAD). Please provide the reference used.
  4. Sample size calculation should be mentioned.
  5. In the statistical analysis, the criteria for selection the variable from the univariable analysis to the multivariable analysis should be shown.

Results

  1. Line 222 and in table 3, univariate and multivariate analyses should be changed to univariable and multivariable analysis.
  2. In all table, the decimal point should be consistency.
  3. In table 2, please recheck the value of % FVC and absolute value of FVC in L.

Are they correct?

  1. In table 3, many factors e.g., AE HOS, UNS are the components of remission. Please clarify these factors are outcomes or history of AE, HOS, UNS in the previous years. If these factors are outcomes, they must be removed from the analysis.
  2. Many factors including BMI, CRS, polyp, and baseline ACT score should be accounted in the multivariable analysis. However, due to small sample size, the model will not strong.
  3. In table 4, the cutoff value of FeNO350 in various cutoff  should be shown e.g., 10, 14, 18, 20,25, etc.

Discussion

  1. There was different in action across the four biologics, some outcomes can be influenced by the differences in the biologic used. This issue should be mentioned in the discussion section as a limitation of the study.

Reviewer 2 Report

Comments and Suggestions for Authors

Dear colleagues 

Thank you for the excellent paper submitted to ARM.

I would like to praise your excellent discussion section on small airway disease, available biomarkers and specific pulmonary testing that are documenting SAD in order to provide precision medicine. 

You have presented in limitation section [page 12 rows 362-367] some aspects that could have influenced your findings. In order to mitigate confusion, I would like to ask you to present briefly the local protocol that is used in your severe asthma center in order to tailor treatment for these patients. How are you deciding that a patient needs a certain approach or another? What is the follow-up for these severe patients?

And chronology of follow-up isn't very clear presented. Because in my country patients with severe Th2-high asthma on biologics have to be reevaluated at 16 weeks from onset of treatment in order to document efficacy. Non-responders have to undergo a complex diagnostic workup and eventually switched on a new biologic. It is not possible to give for one year in a row a biologic if there are no documented benefit at 16 weeks. 

Maybe you could rephrase this statement in page 2 rows 64-66 "“Stable lung function” refers to maintaining forced expiratory volume in one second (FEV1) within ±5% or ±10% of baseline values over a 12-month period, without significant exacerbations or the need for systemic corticosteroids", by deleting the numbers (within ±5% or ±10% of baseline values), because there is no such statement in cited Delphi consensus (Canonica GW, et al. Severe Asthma Network Italy Definition of Clinical Remission in Severe Asthma: A Delphi Consensus. J Allergy Clin Immunol Pract. 2023 Dec;11(12):3629-3637. doi: 10.1016/j.jaip.2023.07.041. Epub 2023 Aug 7. PMID: 37558162.).

Cited paper [reference 6] has 15 times included the terms "pulmonary stability" AND "stability of function" but the only inserted values are present in this statement "Regarding an improvement in lung function as a criterion for remission, there was no consensus on the value to be used as a reference (not 100 or 200 mL, or function greater than or equal to 80% over the uncontrolled phase)", that is 3 times inserted in this text.

If there are other relevant papers that document a certain percentage that is confirming "stable lung function" please rephrase the statement included in rows 64-66 and insert these papers in references list.

Same comments as above for statement in page 3 rows 120-122 "(4) stable lung function, defined as maintaining FEV1 within ±5% or ±10% of baseline values over a 12-month period without the need for systemic corticosteroids."

Eventually you can use only "±10% of baseline values" because it includes lower values of variation like ±5% of baseline if there are available these numbers in other papers than cited paper in reference 6.

It is increasingly evident that severe asthma is not a single disease as evidenced by the variety of clinical presentations, physiologic characteristics and outcomes. To better understand this heterogeneity, the concept of asthma phenotyping has emerged. Please clarify in a more accurate way inclusion criteria of these 25 severe asthma patients presented in current study and generate new tables to validate treatment option for one of the 3 used monoclonal antibodies. A clarification is needed to understand how were allocated patients to one or another of these monoclonals?

Th2-high inflammation phenotype patients could benefit of such a personalized treatment with the listed monoclonals but in table 2 page 7-8 there are values that seem to exclude this particular severe asthma phenotype for some patients with normal eosinophils (in absolute values) and normal IgE titers. Non-responders seem to have late-onset disease (not childhood asthma), longer duration of severe asthma, increased BMI and a female predominance.

Regarding IgE please clarify why are data missing for T1 evaluation? It is imperative for the accuracy of your research to retrieve these missing data.

Why are there no patients treated with omalizumab in your study? 

In row 373 page 12 it is advisable to have a title "Conclusions" and then you insert your statement regarding potential role of  FeNO 350 mL/s ≥18 ppb as response predictor in these asthma patients.

Figure 2 page 10 is a nice artistic achievement, but my humble opinion it isn't suitable for an academic journal of this magnitude. You can use it in a potential video-abstract of the paper when your research will be published.

Some references are incomplete, and pages are missing as in numbers 1, 5, 11, 15, 23, 24, 25

Reference 18 has no data about the journal or site where it was retrieved. 

Reference 9, page 13 row 409 has a significant typo error with a huge empty space, and authors are listed also with their credentials (MD, MBBS, MMedSci, etc.) - has to be corrected 

It would be of great value to unify citation style for all references.

Comments on the Quality of English Language

There are some minor typo errors like

" Follow-up evaluations wew conducted" in row 26 page 1 or

"...within the mounth to mantain a steady expiratory flow" in page 4 row 142

that need to be addressed. 

Positioning in columns seems to be subject of repositioning and term "Cortison" corrected in Table 2 section "Average Annual Cortison Dosage".

Round 2

Reviewer 1 Report

Comments and Suggestions for Authors

All of my comments properly addressed by authors. This mmanuscript an be accepted for publication.

Author Response

We sincerely thank the Reviewer for the positive feedback and for acknowledging our efforts to address all the comments and suggestions. We are truly grateful for the constructive input provided during the review process, which has significantly improved the quality and clarity of our manuscript.

Reviewer 2 Report

Comments and Suggestions for Authors

Dear colleagues I have scrutinized your responses that are providing a significant amount of elucidation for my queries and a major improvement in paper structure and content. Significant explanations were inserted and cleared issues regarding Delphi consensus on stable lung function statement. I praise your efforts to provide new references of real-world data (7-9).

Regarding your explanation in rows 92-97 that is aiming to explain total absence of omalizumab treated patients I still find very unusual that a whole group of omalizumab patients were unable or unwilling to perform "a full set of pulmonary function tests, including fractional exhaled nitric oxide at 350 mL/s (FeNO350)" as stated in rows 94-95 of revised version.

I am encouraging you to insert an extensive section of inclusion-failure patients [with all biologic medication types] because published data on biologic treatment in children across Europe (37 centres in 26 countries) [Santos-Valente E, et al - M. Biologicals in childhood severe asthma: the European PERMEABLE survey on the status quo. ERJ Open Res. 2021 Aug 16;7(3):00143-2021. doi: 10.1183/23120541.00143-2021. PMID: 34409097; PMCID: PMC8365152.] is documenting that all centres have experience and use omalizumab as the most frequent option for first-line treatment in Th2-high severe paediatric asthma patients.

My point is that it is odd to find in real-life cohorts, of any treatment, for any disease, a group of patients that are failing "in corpore" inclusion criteria, for the most commonly used treatment option, at a given time. How often is used in your centre omalizumab? Could help a table or description of all patients out of which you selected the 25 that complied with inclusion criteria and a clear failure rate that was documented.

Statement in row 222 "The prescription of each biologic was performed according to the official criteria of the Italian Medicines Agency (AIFA)." needs a reference number to a specific paper that will provide clear criteria for selecting one or another treatment for severe asthma patients in your country.

Congratulations for the great job performed in references section - still could be improved (see previous comments).

I am yet not concurring with your point of view regarding figure 2 in page 11, but it is the Editor's role to decide.

I am tremendously contented by accuracy and speed of your first revision and I really hope to see these minor adjustments implemented, for the final version.

Author Response

We are sincerely grateful to Reviewer 2 for the careful re-examination of our manuscript and for the constructive feedback that has greatly helped us to improve the clarity, transparency, and overall quality of the paper. We truly appreciate the kind words regarding the accuracy and speed of our first revision and the acknowledgment of the improvements made in structure, references, and methodological explanations.

Dear colleagues I have scrutinized your responses that are providing a significant amount of elucidation for my queries and a major improvement in paper structure and content. Significant explanations were inserted and cleared issues regarding Delphi consensus on stable lung function statement. I praise your efforts to provide new references of real-world data (7-9).

Regarding your explanation in rows 92-97 that is aiming to explain total absence of omalizumab treated patients I still find very unusual that a whole group of omalizumab patients were unable or unwilling to perform "a full set of pulmonary function tests, including fractional exhaled nitric oxide at 350 mL/s (FeNO350)" as stated in rows 94-95 of revised version.

I am encouraging you to insert an extensive section of inclusion-failure patients [with all biologic medication types] because published data on biologic treatment in children across Europe (37 centres in 26 countries) [Santos-Valente E, et al - M. Biologicals in childhood severe asthma: the European PERMEABLE survey on the status quo. ERJ Open Res. 2021 Aug 16;7(3):00143-2021. doi: 10.1183/23120541.00143-2021. PMID: 34409097; PMCID: PMC8365152.] is documenting that all centres have experience and use omalizumab as the most frequent option for first-line treatment in Th2-high severe paediatric asthma patients.

My point is that it is odd to find in real-life cohorts, of any treatment, for any disease, a group of patients that are failing "in corpore" inclusion criteria, for the most commonly used treatment option, at a given time. How often is used in your centre omalizumab? Could help a table or description of all patients out of which you selected the 25 that complied with inclusion criteria and a clear failure rate that was documented.

We thank the Reviewer for this constructive comment. We agree that the absence of omalizumab patients in our final analysis required clarification. As discussed, this likely reflects the characteristics of our adult population, compared with the cited European PERMEABLE survey that also included children and adolescents in whom omalizumab remains the most frequent first-line biologic. In our center, during the 18-month enrollment period, most patients with an allergic phenotype also presented with concomitant eosinophilia, which led to the prescription of anti–IL-5/IL-5R or anti–IL-4R biologics instead of omalizumab. Consequently, only two omalizumab-treated patients were screened and both were excluded due to missing FeNO350 or incomplete follow-up.

To improve transparency, we have:

  1. Added Supplementary Table S1, reporting all excluded patients (biologic therapy, age, sex, and reason for exclusion).
  2. Integrated the Methods section with a new paragraph placed at the beginning of subsection 2.2 Data Collection, clearly describing the screening denominator and the reasons for exclusion.

We believe these additions address the Reviewer’s concern and strengthen the methodological clarity of our manuscript.

Statement in row 222 "The prescription of each biologic was performed according to the official criteria of the Italian Medicines Agency (AIFA)." needs a reference number to a specific paper that will provide clear criteria for selecting one or another treatment for severe asthma patients in your country.

We thank the Reviewer for this important remark. We have now added a formal citation to the Italian Medicines Agency (AIFA), which publishes the official therapeutic plans and eligibility criteria for the prescription of biologics in severe asthma. The sentence in row 222 has been revised as follows:

“The prescription of each biologic was performed according to the official criteria of the Italian Medicines Agency (AIFA) [18]”Accordingly, the AIFA website has been inserted into the References list

Congratulations for the great job performed in references section - still could be improved (see previous comments).

I am yet not concurring with your point of view regarding figure 2 in page 11, but it is the Editor's role to decide.

I am tremendously contented by accuracy and speed of your first revision and I really hope to see these minor adjustments implemented, for the final version.

Round 3

Reviewer 2 Report

Comments and Suggestions for Authors

Thank you for your patience and resilience dear colleagues! Your comments and explanation regarding monoclonals usage in your center is extremely relevant.

Also, addition of table S1 in Supplementary files and reference 18 (Italian Medicines Agency (AIFA). Criteria for Prescribing Biologic Therapies in Severe Asthma. Available online: https://www.aifa.gov.it) enhances significantly potential impact of your important research and clarified my previous interrogations. 

Thank you again and success!